# Determining Key Agricultural Strategic Factors Using AHP-MICMAC

**Ali Akbar Barati [1],\*** , **Hossein Azadi [2,3]** , **Milad Dehghani Pour [1], Philippe Lebailly [4] and Mostafa Qafori [1]**

[1]  Department of Agricultural Management and Development, University of Tehran, Tehran, Iran
[2]  Department of Geography, Ghent University, Ghent 9000, Belgium
[3]  ISUMADECIP, Faculty of Environmental Science and Engineering, Babeş-Bolyai University, Cluj-Napoca 400084, Romania
[4]  Department of Economics and Rural Development, Gembloux Agro-Bio Tech, University of Liège, 5030 Gembloux, Belgium
\*  Correspondence: aabarati@ut.ac.ir

**Abstract:** Agriculture is an irrefutable part of food policy. This paper aims to introduce an integrated method using MICMAC and AHP techniques to deal with understanding the key strategic variables of agricultural system. MICMAC was used to determine the classifications of variables and AHP was applied to weigh these classifications. MICMAC is a structural analysis tool used to structure ideas and AHP is an effective tool to deal with complex decision making and helps decision-makers making the best decision. The results show that strategic variables had different types of influence and direct, indirect, and potential dependencies did not have the same importance. AHP-MICMAC not only considers these differences, but also puts a total priority weight for each variable. These characteristics have an important role in forming strategies and scenarios for agricultural development. Therefore, the case of Iran was used to illustrate the application of MICMAC aiming to supply instructions for the development of agriculture system.

**Keywords:** agricultural policy; agriculture management; policy-making for food; decision making for agriculture; Iran agriculture

## 1. Introduction

Agriculture is a complex system [1], but due to the risky and diverse nature of agriculture in developing countries, the systemic complexity is greater [2]. In a complex system, there are a variety of autonomous actors, just as a variety of actors and processes of adaptation can be found within the agricultural system [3]: humans (farmers, laborers, consumers, policy makers, experts, agents, etc.) [4,5], economy (market, cost, income, etc.) [6], nature (weather and climate, topology, etc.) [7–9], policy (plans, policies, strategies, etc.) [10], regulations (heritage, property rights, trade, etc.) [11], infrastructures (transportation, processing, saving, marketing, insurance, etc.) [12–14], inputs (land, water, seed, fertilizer, technology, etc.) [5,15–17].

Determining what kind of factors or variables need to be considered by decision and policy-makers is challenging [18]. Policy-makers tend to use different criteria and methodologies in order to determine strategic variables and factors influencing agricultural development [19]. Yet, because of the complexity of agricultural systems, the ability of researchers and policy-makers to prioritize variables is often limited. As a result, the majority of previous studies have dealt with this subject from a limited point of view (such as insurance or risk management) and on a micro level (such as a single farmer or farm). For instance, Pascucci and de-Magistris [20] implemented a multivariate probit model to

evaluate the effects of different types of agricultural extension and innovation systems on farmers' strategies in Italy. Allen [21] used the bet-hedging model and Neo-Darwinian theory (risk management strategies) to offer a way of evaluating the historical development of dryland agriculture as well as the long-term outcomes of variant agronomic strategies in Kona, Hawaii. Qingshui and Xuewei [22] and Zhou et al. [23] used empirical research to develop and improve strategies for the agricultural insurance system in rural of China by considering income sources, mean of production, labor opportunities, government supports, and communication channels. In Anambra, Nigeria, Amadi [14] evaluated the impact of rural road construction and its adjacent infrastructures (electricity, pipe-borne water and irrigation technology) that were used as a strategy for rural and agricultural development. Ames [4] emphasized investment in human capital as a strategy for implementing changes in agricultural policy, research, and extension activities.

Most of these studies only considered a few limiting factors or variables and their intensities, but none of them attended to characteristics such as dependent or independent variables, direct or indirect impacts, or the weight of each variable or factor. These characteristics have an important role in forming strategies and scenarios for agricultural development. As a result, there is a methodological gap that the present study aims to fill by providing a new integrated method. This new integrated method applies Impact Matrix Cross-reference Multiplication to a classification (MICMAC) [24,25] and analytic hierarchy process (AHP) [26]. The case of the agricultural system in Iran is used to show the application of this new methodology. Agriculture is one of the most important sectors of the Iranian economy, accounting for about 11% of GDP, 23% of the employed population, and 15% of the foreign exchange revenue (form non-oil exports). In addition to the fact that products from the agriculture and animal husbandry have been major export commodities, including pistachios, raisins, and even carpets. About 20% of Iran is arable, with some northern and western areas that support rain-fed agriculture, while other areas require irrigation.

Each of these methods alone has advantages and limitations for example MICMAC can investigate multiple variables at the same time, but it does not give an overall priority score for each variable. On other side, AHP considers only direct impact of variables, but it gives and overall priority score for each variable. This study has tried to overcome these constraints and to consider their advantages by combining them and proposing an integrated method. It is our hope that this new integrated method will supply instructions for the development of agriculture, and find wider applications in complex systems.

## 2. Materials and Methods

### 2.1. MICMAC Method

The Impact Matrix Cross-Reference Multiplication Applied to a Classification (MICMAC) is a structural analysis tool used to structure ideas and as a forecasting method created by Michel Godet. MICMAC can be considered a qualitative system dynamics approach [27] and provides the possibility to describe a system with the help of a matrix connecting all its components. By studying these relations, the method also makes it possible to reveal the variables essential to the evolution of the system. It is possible to use MICMAC as an aid for reflection and/or for decision making, or as a part of a more complex forecasting activity [28]. MICMAC tries to pinpoint the independent and dependent variables by building a typology in both direct and indirect classifications [28]. In MICMAC we depart from the definition of the system's variables and their interrelations, both of which were provided by experts. This method has at least three main phases [25,28,29]:

Phase (1) Considering all the variables: This phase begins by considering all of the variables or factors that characterize the studied system. Brainstorming and intuitive methods or a panel of experts are useful methods for this phase. A detailed explanation of the variables is also essential because it will allow the relations between these variables to be perceived better in the analysis. The final output

of this phase is a homogeneous list of internal and external variables (Table 1) and should not exceed more than 70 to 80 variables.

**Table 1.** A sample list of internal and external variables.

| Type of Variables | Group Name (if it is necessary) | No | Symbol | Description (Lable) |
|---|---|---|---|---|
| Internal variables | Economic | 1 | Var1 | Income . . . |
| | | 2 | Var2 | . . . |
| | Social | 3 | Var3 | Participation . . . |
| | | 4 | Var4 | . . . |
| External variables | . . . | 5 | Var5 | . . . |
| | | 6 | Var6 | . . . |
| | . . . | 7 | Var7 | . . . |
| | | 8 | Var8 | . . . |

*Phase (2) Constructing the structural analysis matrix (description of the relations between the variables):* In a systemic vision, a variable is a part of the relational web. A structural analysis matrix is a squared matrix that allows the variables to connect directly. The cells store the degree of influence between each pair of variables, *i* and *j* (0 no influence, 1 weak influence, 2 medium, 3 strong and P potential) (Table 2) (A group of experts filled this matrix). This filling-in phase helps place $N \times (N-1)$ questions for N variables. Additionally, the questioning procedure not only enables us to avoid errors, but also helps us organize and classify ideas by creating a common group language. It also allows for the variables to be redefined and therefore makes analysis of the system more accurate. Figure 1 indicates the structural diagram of Table 2.

**Table 2.** A sample structural analysis matrix (M) with four variables.

| Variables | Var1 | Var2 | Var3 | Var4 | Influence |
|---|---|---|---|---|---|
| Var1 | 0 | 0 | 1 | 3 | 4 |
| Var2 | 1 | P | 1 | 0 | 2 |
| Var3 | 0 | 2 | 0 | 0 | 2 |
| Var4 | 0 | 1 | 3 | 0 | 4 |
| Dependence | 1 | 3 | 5 | 3 | - |

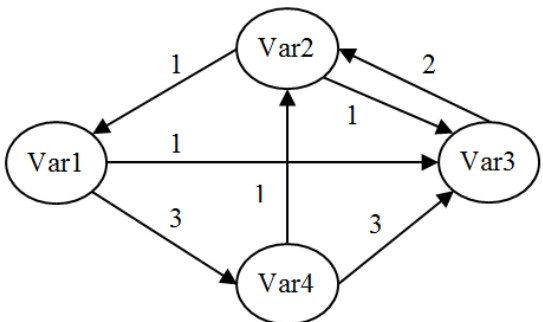

**Figure 1.** A structural diagram (based on the data of Table 2).

*Phase (3) Identification of the key variables:* This phase consists of identifying variables essential to the system's development. At first, this was accomplished by using direct classification, then through indirect classification and, finally, by potential classification. Comparing the hierarchy of variables in the various types of classifications (direct, indirect, and potential) is a rich source of information. It enables us not only to confirm the importance of certain variables, but also to uncover variables which play an important role yet were not identifiable through direct classification in the initial process.

The direct influence and dependence of a variable are the aggregate of its row and column. The sum of each row indicates the importance of the influence of a variable on the whole system (other variables) (Equation (1)) and the sum of a column indicates the degree of dependence of a variable on the other variables (Equation (2)):

$$DI_j = \sum_{j=1}^{n} V_{ij} \ i \text{ and } j \ = \ 1, 2, \ldots, n \tag{1}$$

$$DD_i = \sum_{i=1}^{n} V_{ij} i \text{ and } j \ = \ 1, 2, \ldots, n \tag{2}$$

Indirect classification is obtained after increasing the power of the matrix M (matrix multiplication $M^2 = M \times M$, $M^3 = M \times M \times M$, and so on). For example, in Figure 1 Var1 has a direct $(DI_{13} = V_1 \rightarrow V_3 = 1)$ and indirect $(II_{13} = V_1 \rightarrow V_4 \rightarrow V_3)$ influence on Var3. To calculate indirect influence or dependence of a path, we should increase the power of the matrix by considering the number of paths and loops of length $(1, 2, \ldots, N)$ that result from or arrive at each variable (for example, for $II_{13} = V_1 \rightarrow V_4 \rightarrow V_3$, the power of the matrix should be Equation (2)). The MICMAC then allows us to study the diffusion of the impacts through the paths and loops of feedback. Generally, the classification becomes stable after a degree of multiplication of 3, 4 or 5 [29].

$$M = \begin{vmatrix} 0 & 0 & 1 & 3 \\ 1 & 0 & 1 & 0 \\ 0 & 2 & 0 & 0 \\ 0 & 1 & 3 & 0 \end{vmatrix} \rightarrow M^2 = \begin{vmatrix} 0 & 5 & 9 & 0 \\ 0 & 2 & 1 & 3 \\ 2 & 0 & 2 & 0 \\ 1 & 6 & 1 & 0 \end{vmatrix}$$

A potential direct or indirect classification is a direct or indirect relation (influence or dependence) that considers potential relations. To calculate potential relations, we ought to first replace P in matrix M with an ordinal number (1, 2, or 3, depending on the intensity of influence) and then increase the power of the new matrix to a point where the row and column priorities become stable. If there is no potential influence or dependence, the degree of potential relations will be equal to existing relations. In simple terms, feedback loops may take a number of iterations to come to a settled state. The number of times that the matrix can be multiplied depends upon how long it takes to stabilize.

MICMAC compared to the results (direct, indirect, and potential classification) provides the possibility to confirm the importance of variables. The main result of this phase is a matrix $m \times n$ (Table 3), which we named matrix R; where $m$ is the number of various types of relations (various types of classifications). Here it includes eight types: Direct Influence (DI), Indirect Influence (II), Direct Dependence (DD), Indirect Dependence (ID), Potential Direct Influences (PDI), Potential Indirect Influence (PII), Potential Direct Dependence (PDD), and Potential Indirect Dependence (PID). $N$ represents the number of variables. A comparison of the hierarchy within the variables provides a rich source of information.

**Table 3.** Identification of the key variables according to various types of classifications (Matrix R).

| Matrix R | DI | II | DD | ID | PDI | PII | PDD | PID |
|----------|----|----|----|----|-----|-----|-----|-----|
| $V_1$ | | | | | | | | |
| $V_2$ | | | | | | | | |
| ⋮ | | | | | | | | |
| $V_n$ | | | | | | | | |

## 2.2. AHP Method

The analytic hierarchy process (AHP) is a structured technique developed by Thomas L. Saaty in the 1970s. It is an effective tool when dealing with complex decision making and helps decision-makers to set priorities and make the best decision. AHP uses a series of paired comparisons to reduce complex decisions. Then, by synthesizing the results, it helps capture both the subjective and objective aspects

of a decision. Additionally, AHP is used to reduce bias in a decision making process and incorporates a useful technique that checks the consistency of the decision-maker's judgments [26,30,31].

The AHP can be implemented through the following steps:

1. Define the problem and determine the objectives, criteria, sub-criteria, and alternatives.
2. Structure the decision hierarchy from the top (the goal of the decision), down (the alternatives).
3. Construct a set of paired comparison matrices. Each element on an upper level is used to compare the elements at the level immediately below it.
4. Compute the vector of criteria weights.
5. Compute the matrix of option scores. For each element in the level below, add its weighed values and obtain its overall or global priority.
6. Rank the options (alternatives).

Each step will be described in detail. We assume that the *m* evaluation criteria are considered as evaluated *n* options or alternatives (in our study, 45 variables).

*(1) Define the problem:* Our problem or goal was determining the strategic variables of agricultural development based on various types of classifications.

*(2) Structure the decision hierarchy:* The structure of our decision hierarchy is shown in Figure 2. This hierarchical process includes three levels: (a) Goal (in our study it was to determine the strategic variables of an agricultural system), (b) criteria (in our study they were eight types of classifications: DI, II, DD, ID, PDI, PII, PDD, and PID), and (c) alternative variables.

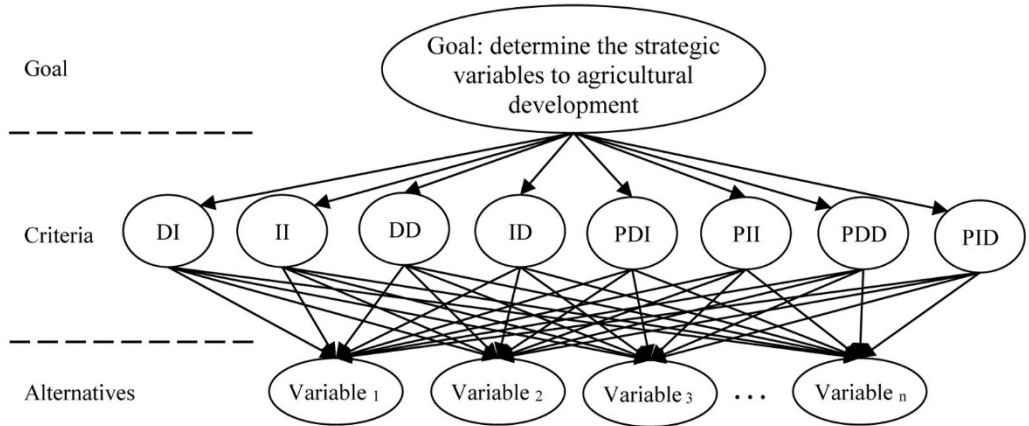

**Figure 2.** The structure of the decision hierarchy.

*(3) Create a paired comparison matrix (A):* Matrix A is a $m \times m$ matrix. Each entry, $a_{ij}$, presents the importance of the *i*th criterion relative to the *j*th criterion. If $a_{ij} = k$ and $k > 1$, it means that the *i*th criterion is *k* times more important than the *j*th criterion, while if $a_{ij} = k$ and $k < 1$, it means that the *i*th criterion is *k* times less important than the *j*th criterion. If $k = 1$, then the two criteria have the same importance. The entries $a_{ij}$ and $a_{ji}$ satisfy this constraint, $a_{ij} \times a_{ji} = 1$ ($a_{ij} = 1/a_{ji}$). The relative importance between two criteria is measured according to a numerical scale, from 1 to 9 (1 for equal importance of *i* and *j*, ... , 9 absolutely *i* is more important than *j*). The consistency index (CI) [31] was used to check the reliability of the paired comparisons.

$$
A = \quad
\begin{array}{c|cccc}
 & a_1 & a_2 & .. & a_j \\
\hline
a_1 & a_{11} & a_{12} & .. & a_{1j} \\
a_2 & a_{21} & a_{22} & .. & a_{2j} \\
a_i & a_{i1} & a_{i2} & .. & a_{ij}
\end{array}
$$

*(4) Compute the vector of criteria weight:* Once matrix A is built, it should be normalized. To this purpose, the sum of the entries on each column should be made equal to 1. In the resulting matrix (A$_{norm}$), each entry $\bar{a}_{ij}$ is computed as (Equation (3)):

$$\bar{a}_{ij} = \frac{a_{ij}}{\sum_{j=1}^{m} a_{ij}} \tag{3}$$

Finally, the criteria weight vector $w$ is built by averaging the entries in each row of matrix A$_{norm}$ (Equation (4)).

$$w_i = \frac{\sum_{i=1}^{m} \bar{a}_{ij}}{m} \tag{4}$$

*(5) Compute the matrix of option scores:* This matrix is a $m \times n$ real matrix (S). Each entry $s_{ij}$ of S represents the score of the *i*th option with respect to the *j*th criterion. In our study this matrix was the output of the MICMAC method and was a $8 \times 45$ matrix (8 types or relations and 45 strategic variables).

*(6) Rank the options or alternatives (variables):* in this phase a vector $v$ of global scores is obtained by multiplying matrix S and vector $w$, i.e., $v_j = S \times w_i$.

## 2.3. AHP-MICMAC Integrated Method

Although the MICMAC method is useful when identifying key variables and it gives us the priority of each variable according to different types of relations (from direct influence to potential indirect dependence), it couldn't calculate a proper weight for the types of relations or an overall priority ranking with respect to these weights of each variable. Thus, we introduced an integrated method (AHP-MICMAC) to deal with this problem. As Figure 3 indicates, AHP-MICMAC can be implemented in eight simple consecutive steps:

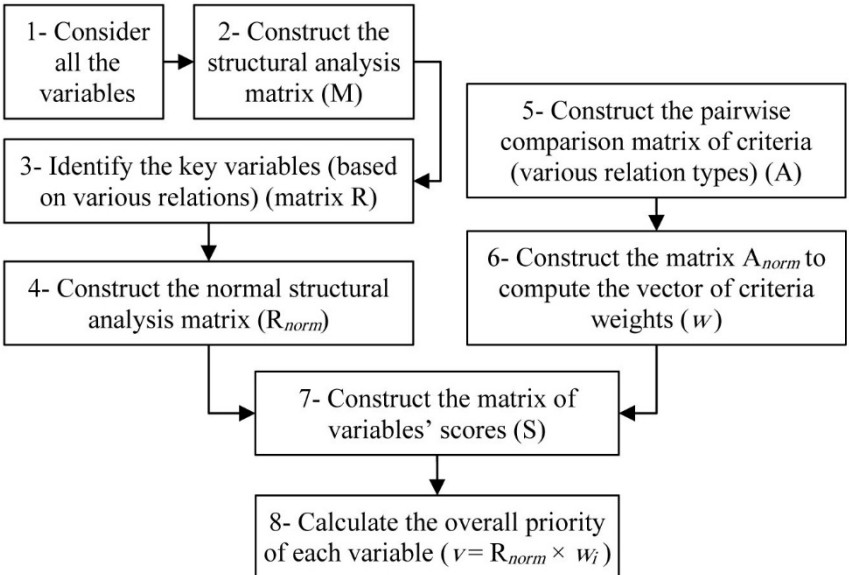

**Figure 3.** The process of the AHP-MICMAC method.

*(1) Consider all the variables:* At first, we prepared a list of important variables extracted from literature review. Then, we organized a panel of 10 experts (including five faculty members of Agricultural Economics and Development at University of Tehran and five experienced experts of Agricultural Ministry) in order to prepare a final list of all variables that are fundamental for the development of agriculture in Iran. Brainstorming among the group, the panel finally extracted 45 variables as the key variables of agricultural development (Table 4).

**Table 4.** The key variables for agricultural development.

| Label | Variable | Label | Variable |
|-------|----------|-------|----------|
| V01 | The demand for agricultural products | V24 | Farmers' knowledge, awareness and skills |
| V02 | Consumers' interest and motivation | V25 | Agricultural extension and education |
| V03 | Consumers' knowledge and awareness | V26 | Agricultural research |
| V04 | Consumers' purchasing power | V27 | The amount of water resources |
| V05 | Consumers' access to agricultural products | V28 | Water efficiency |
| V06 | Marketing | V29 | Climate (temperature and precipitation) |
| V07 | A proper network of markets | V30 | Technology |
| V08 | A price system and pricing | V31 | Agricultural support system |
| V09 | Transportation and communications | V32 | Agricultural land area |
| V10 | Data and Information network | V33 | Agricultural land laws and regulations |
| V11 | Processing and packaging | V34 | Soil texture |
| V12 | Agricultural products price | V35 | Topology |
| V13 | A system for consumer protection | V36 | Land fragmentation |
| V14 | Commercial infrastructure | V37 | Optimum use of inputs (seeds, fertilizers, etc.) |
| V15 | Storage facilities | V38 | The price of production inputs |
| V16 | Trade incentives and restrictions | V39 | Existence of rural job opportunities |
| V17 | The amount of agricultural production | V40 | The number of agricultural labor |
| V18 | Production costs | V41 | The international prices of agricultural products |
| V19 | Government policies and programs | V42 | Involving farmers in agricultural development |
| V20 | Quality agricultural products | V43 | Farmers organizing and institutionalizing |
| V21 | Farmers' interest and motivation | V44 | Disasters (droughts, floods, earthquakes, etc.) |
| V22 | Financial ability of farmers | V45 | Crop insurance |
| V23 | Rural welfare and comforting | | |

*(2) Construct the structural analysis matrix (M):* We constructed a $45 \times 45$ matrix of key variables and asked a panel of experts to score the degree of influence between each pair of variables on a scale from 0 to 3 (0 no influence; 1, weak influence; 2, medium influence; and 3, strong influence) (Table 5).

**Table 5.** A part of the constructed structural analysis matrix (M).

| M | V01 | V02 | V03 | V04 | V05 | V06 | V07 | V08 | V09 | V10 | V11 | V12 | V13 | V14 | V15 |
|-----|-----|-----|-----|-----|-----|-----|-----|-----|-----|-----|-----|-----|-----|-----|-----|
| **V01** | 0 | 0 | 0 | 0 | 0 | 3 | 3 | 2 | 2 | 0 | 2 | 3 | 2 | 2 | 2 |
| **V02** | 3 | 0 | 2 | 0 | 0 | 1 | 0 | 0 | 0 | 0 | 2 | 2 | 0 | 1 | 0 |
| **V03** | 2 | 2 | 0 | 0 | 0 | 0 | 0 | 0 | 0 | 0 | 2 | 0 | 0 | 0 | 0 |
| **V04** | 3 | 2 | 1 | 0 | 2 | 0 | 0 | 2 | 0 | 0 | 0 | 2 | 3 | 0 | 0 |
| **V05** | 2 | 1 | 0 | 0 | 0 | 0 | 0 | 0 | 0 | 0 | 0 | 0 | 1 | 0 | 0 |
| **V06** | 3 | 2 | 1 | 0 | 2 | 0 | 3 | 2 | 2 | 2 | 3 | 3 | 0 | 2 | 3 |
| **V07** | 3 | 0 | 0 | 0 | 3 | 3 | 0 | 0 | 2 | 1 | 0 | 0 | 2 | 0 | 2 |
| **V08** | 3 | 2 | 0 | 0 | 0 | 2 | 0 | 0 | 0 | 0 | 0 | 1 | 3 | 0 | 0 |
| **V09** | 1 | 0 | 0 | 0 | 3 | 2 | 2 | 0 | 0 | 0 | 2 | 2 | 0 | 2 | 1 |
| **V10** | 2 | 2 | 2 | 0 | 1 | 2 | 0 | 0 | 0 | 0 | 0 | 1 | 0 | 0 | 0 |
| **V11** | 3 | 2 | 0 | 0 | 1 | 3 | 0 | 0 | 0 | 0 | 0 | 2 | 0 | 0 | 2 |
| **V12** | 3 | 2 | 0 | 2 | 0 | 2 | 2 | 3 | 0 | 0 | 2 | 0 | 2 | 0 | 2 |
| **V13** | 3 | 3 | 2 | 2 | 2 | 1 | 0 | 1 | 0 | 0 | 0 | 2 | 0 | 1 | 2 |
| **V14** | 0 | 0 | 0 | 0 | 2 | 2 | 2 | 1 | 2 | 1 | 3 | 2 | 0 | 0 | 3 |
| **V15** | 0 | 0 | 0 | 0 | 2 | 2 | 0 | 2 | 0 | 0 | 2 | 2 | 0 | 0 | 0 |

*(3) Identify the key variables (based of various relations) (matrix R):* Using MICMAC software (Version 6.1.2 [32]), we identified the key variables based on 8 different types of relations: Direct Influence (DI), Indirect Influence (II), Direct Dependence (DD), Indirect Dependence (ID), Potential Direct Influences (PDI), Potential Indirect Influence (PII), Potential Direct Dependence (PDD), and Potential Indirect Dependence (PID) (Tables 3 and 6).

*(4) Construct the normal structural analysis matrix ($R_{norm}$):* During this phase, Equation (3) was applied to matrix R to convert to matrix $R_{norm}$ (Table 6).

**Table 6.** The key variables based on various relations (matrix R and matrix R*norm*).

| Matrix | Non-Normal (Matrix R) | | | | | | | | Normal (Matrix R*norm*) | | | | | | | |
|---|---|---|---|---|---|---|---|---|---|---|---|---|---|---|---|---|
| | DI | II | DD | ID | PDI | PII | PDD | PID | DI | II | DD | ID | PDI | PII | PDD | PID |
| V01 | 243 | 237 | 314 | 323 | 243 | 239 | 314 | 321 | 0.024 | 0.024 | 0.031 | 0.032 | 0.024 | 0.024 | 0.031 | 0.032 |
| V02 | 114 | 122 | 171 | 176 | 114 | 123 | 171 | 174 | 0.011 | 0.012 | 0.017 | 0.018 | 0.011 | 0.012 | 0.017 | 0.017 |
| V03 | 71 | 67 | 64 | 51 | 71 | 67 | 64 | 50 | 0.007 | 0.007 | 0.006 | 0.005 | 0.007 | 0.007 | 0.006 | 0.005 |
| V04 | 157 | 146 | 64 | 80 | 157 | 147 | 64 | 78 | 0.016 | 0.015 | 0.006 | 0.008 | 0.016 | 0.015 | 0.006 | 0.008 |
| V05 | 64 | 65 | 171 | 176 | 64 | 65 | 171 | 174 | 0.006 | 0.007 | 0.017 | 0.018 | 0.006 | 0.007 | 0.017 | 0.017 |
| V06 | 386 | 361 | 336 | 382 | 386 | 363 | 336 | 381 | 0.039 | 0.036 | 0.034 | 0.038 | 0.039 | 0.036 | 0.034 | 0.038 |
| V07 | 200 | 193 | 143 | 166 | 200 | 194 | 143 | 166 | 0.020 | 0.019 | 0.014 | 0.017 | 0.020 | 0.019 | 0.014 | 0.017 |
| V08 | 164 | 184 | 236 | 270 | 164 | 184 | 236 | 269 | 0.016 | 0.018 | 0.024 | 0.027 | 0.016 | 0.018 | 0.024 | 0.027 |
| V09 | 200 | 183 | 193 | 186 | 200 | 183 | 193 | 185 | 0.020 | 0.018 | 0.019 | 0.019 | 0.020 | 0.018 | 0.019 | 0.019 |
| V10 | 264 | 265 | 121 | 123 | 264 | 264 | 121 | 123 | 0.026 | 0.027 | 0.012 | 0.012 | 0.026 | 0.026 | 0.012 | 0.012 |
| V11 | 250 | 228 | 343 | 379 | 250 | 230 | 343 | 377 | 0.025 | 0.023 | 0.034 | 0.038 | 0.025 | 0.023 | 0.034 | 0.038 |
| V12 | 343 | 325 | 450 | 478 | 343 | 325 | 450 | 478 | 0.034 | 0.033 | 0.045 | 0.048 | 0.034 | 0.033 | 0.045 | 0.048 |
| V13 | 257 | 236 | 135 | 150 | 257 | 237 | 135 | 149 | 0.026 | 0.024 | 0.014 | 0.015 | 0.026 | 0.024 | 0.014 | 0.015 |
| V14 | 271 | 254 | 114 | 121 | 271 | 255 | 114 | 120 | 0.027 | 0.025 | 0.011 | 0.012 | 0.027 | 0.026 | 0.011 | 0.012 |
| V15 | 178 | 181 | 278 | 320 | 178 | 181 | 278 | 318 | 0.018 | 0.018 | 0.028 | 0.032 | 0.018 | 0.018 | 0.028 | 0.032 |
| V16 | 250 | 237 | 185 | 196 | 250 | 238 | 185 | 195 | 0.025 | 0.024 | 0.019 | 0.020 | 0.025 | 0.024 | 0.019 | 0.020 |
| V17 | 336 | 320 | 672 | 578 | 336 | 321 | 672 | 582 | 0.034 | 0.032 | 0.067 | 0.058 | 0.034 | 0.032 | 0.067 | 0.058 |
| V18 | 271 | 275 | 436 | 389 | 271 | 275 | 436 | 393 | 0.027 | 0.028 | 0.044 | 0.039 | 0.027 | 0.028 | 0.044 | 0.039 |
| V19 | 565 | 505 | 472 | 404 | 565 | 507 | 472 | 408 | 0.057 | 0.051 | 0.047 | 0.040 | 0.057 | 0.051 | 0.047 | 0.041 |
| V20 | 243 | 223 | 393 | 336 | 243 | 224 | 393 | 339 | 0.024 | 0.022 | 0.039 | 0.034 | 0.024 | 0.022 | 0.039 | 0.034 |
| V21 | 128 | 151 | 436 | 385 | 128 | 150 | 436 | 388 | 0.013 | 0.015 | 0.044 | 0.039 | 0.013 | 0.015 | 0.044 | 0.039 |
| V22 | 271 | 263 | 429 | 422 | 271 | 263 | 429 | 422 | 0.027 | 0.026 | 0.043 | 0.042 | 0.027 | 0.026 | 0.043 | 0.042 |
| V23 | 193 | 188 | 207 | 201 | 193 | 186 | 207 | 202 | 0.019 | 0.019 | 0.021 | 0.020 | 0.019 | 0.019 | 0.021 | 0.020 |
| V24 | 336 | 335 | 121 | 135 | 336 | 334 | 121 | 134 | 0.034 | 0.034 | 0.012 | 0.014 | 0.034 | 0.033 | 0.012 | 0.013 |
| V25 | 286 | 279 | 264 | 285 | 286 | 279 | 264 | 283 | 0.029 | 0.028 | 0.026 | 0.029 | 0.029 | 0.028 | 0.026 | 0.028 |
| V26 | 214 | 254 | 193 | 240 | 214 | 253 | 193 | 238 | 0.021 | 0.025 | 0.019 | 0.024 | 0.021 | 0.025 | 0.019 | 0.024 |
| V27 | 243 | 259 | 157 | 150 | 243 | 258 | 157 | 150 | 0.024 | 0.026 | 0.016 | 0.015 | 0.024 | 0.026 | 0.016 | 0.015 |
| V28 | 214 | 231 | 228 | 224 | 214 | 230 | 228 | 225 | 0.021 | 0.023 | 0.023 | 0.022 | 0.021 | 0.023 | 0.023 | 0.023 |
| V29 | 243 | 251 | 42 | 43 | 243 | 250 | 42 | 43 | 0.024 | 0.025 | 0.004 | 0.004 | 0.024 | 0.025 | 0.004 | 0.004 |
| V30 | 393 | 372 | 264 | 265 | 393 | 373 | 264 | 264 | 0.039 | 0.037 | 0.026 | 0.027 | 0.039 | 0.037 | 0.026 | 0.026 |

**Table 6.** *Cont.*

| Matrix | Non-Normal (Matrix R) | | | | | | | | Normal (Matrix R*norm*) | | | | | | | |
|---|---|---|---|---|---|---|---|---|---|---|---|---|---|---|---|---|
| | DI | II | DD | ID | PDI | PII | PDD | PID | DI | II | DD | ID | PDI | PII | PDD | PID |
| V31 | 250 | 274 | 185 | 195 | 250 | 273 | 185 | 195 | 0.025 | 0.027 | 0.019 | 0.020 | 0.025 | 0.027 | 0.019 | 0.020 |
| V32 | 200 | 209 | 243 | 215 | 200 | 208 | 243 | 218 | 0.020 | 0.021 | 0.024 | 0.022 | 0.020 | 0.021 | 0.024 | 0.022 |
| V33 | 157 | 181 | 114 | 109 | 157 | 180 | 114 | 109 | 0.016 | 0.018 | 0.011 | 0.011 | 0.016 | 0.018 | 0.011 | 0.011 |
| V34 | 128 | 135 | 135 | 115 | 128 | 134 | 135 | 116 | 0.013 | 0.014 | 0.014 | 0.012 | 0.013 | 0.013 | 0.014 | 0.012 |
| V35 | 114 | 121 | 0 | 0 | 114 | 122 | 0 | 0 | 0.011 | 0.012 | 0.000 | 0.000 | 0.011 | 0.012 | 0.000 | 0.000 |
| V36 | 135 | 146 | 85 | 86 | 135 | 145 | 85 | 86 | 0.014 | 0.015 | 0.009 | 0.009 | 0.014 | 0.015 | 0.009 | 0.009 |
| V37 | 150 | 152 | 243 | 199 | 150 | 152 | 243 | 202 | 0.015 | 0.015 | 0.024 | 0.020 | 0.015 | 0.015 | 0.024 | 0.020 |
| V38 | 114 | 134 | 150 | 166 | 114 | 133 | 150 | 165 | 0.011 | 0.013 | 0.015 | 0.017 | 0.011 | 0.013 | 0.015 | 0.017 |
| V39 | 85 | 73 | 128 | 178 | 85 | 72 | 128 | 177 | 0.009 | 0.007 | 0.013 | 0.018 | 0.009 | 0.007 | 0.013 | 0.018 |
| V40 | 92 | 109 | 200 | 249 | 92 | 108 | 200 | 248 | 0.009 | 0.011 | 0.020 | 0.025 | 0.009 | 0.011 | 0.020 | 0.025 |
| V41 | 85 | 97 | 185 | 194 | 85 | 97 | 185 | 193 | 0.009 | 0.010 | 0.019 | 0.019 | 0.009 | 0.010 | 0.019 | 0.019 |
| V42 | 228 | 247 | 135 | 143 | 228 | 246 | 135 | 142 | 0.023 | 0.025 | 0.014 | 0.014 | 0.023 | 0.025 | 0.014 | 0.014 |
| V43 | 371 | 390 | 207 | 199 | 371 | 390 | 207 | 198 | 0.037 | 0.039 | 0.021 | 0.020 | 0.037 | 0.039 | 0.021 | 0.020 |
| V44 | 357 | 348 | 35 | 3 | 357 | 348 | 35 | 4 | 0.036 | 0.035 | 0.004 | 0.000 | 0.036 | 0.035 | 0.004 | 0.000 |
| V45 | 164 | 174 | 300 | 294 | 164 | 173 | 300 | 293 | 0.016 | 0.017 | 0.030 | 0.029 | 0.016 | 0.017 | 0.030 | 0.029 |
| Sum | 9978 | 9980 | 9977 | 9979 | 9978 | 9979 | 9977 | 9975 | 1 | 1 | 1 | 1 | 1 | 1 | 1 | 1 |

*(5) Construct the paired comparison matrix of criteria (A):* Since the MICMAC method includes eight different types of classifications (DI, II, DD, ID, PDI, PII, PDD and PID), there are eight criteria. Therefore, the paired comparison matrix A is an 8 × 8 matrix. The following matrix is the constructed matrix A for this study:

|  |  | DI | II | DD | ID | PDI | PII | PDD | PID |
|---|---|---|---|---|---|---|---|---|---|
|  | **DI** | 1.00 | 2.00 | 2.00 | 4.00 | 2.00 | 4.00 | 4.00 | 8.00 |
|  | **II** | 0.50 | 1.00 | 1.00 | 2.00 | 1.00 | 2.00 | 2.00 | 4.00 |
|  | **DD** | 0.50 | 1.00 | 1.00 | 2.00 | 1.00 | 2.00 | 2.00 | 4.00 |
| A= | **ID** | 0.25 | 0.50 | 0.50 | 1.00 | 0.50 | 1.00 | 1.00 | 2.00 |
|  | **PDI** | 0.50 | 1.00 | 1.00 | 2.00 | 1.00 | 2.00 | 2.00 | 4.00 |
|  | **PII** | 0.25 | 0.50 | 0.50 | 1.00 | 0.50 | 1.00 | 1.00 | 2.00 |
|  | **PDD** | 0.25 | 0.50 | 0.50 | 1.00 | 0.50 | 1.00 | 1.00 | 2.00 |
|  | **PID** | 0.13 | 0.25 | 0.25 | 0.50 | 0.25 | 0.50 | 0.50 | 1.00 |

*(6) Construct the matrix $A_{norm}$ to compute the vector of criteria weights (w)*: The matrix $A_{norm}$ and the vector of criteria weights (w) were calculated, respectively, using Equations (3) and (4). The matrix and vector for our study are indicated below:

|  |  | DI | II | DD | ID | PDI | PII | PDD | PID | $w_i$ |
|---|---|---|---|---|---|---|---|---|---|---|
|  | **DI** | 0.296 | 0.296 | 0.296 | 0.296 | 0.296 | 0.296 | 0.296 | 0.296 | 0.296 |
|  | **II** | 0.148 | 0.148 | 0.148 | 0.148 | 0.148 | 0.148 | 0.148 | 0.148 | 0.148 |
|  | **DD** | 0.148 | 0.148 | 0.148 | 0.148 | 0.148 | 0.148 | 0.148 | 0.148 | 0.148 |
| $A_{norm}$ = | **ID** | 0.074 | 0.074 | 0.074 | 0.074 | 0.074 | 0.074 | 0.074 | 0.074 | 0.074 |
|  | **PDI** | 0.148 | 0.148 | 0.148 | 0.148 | 0.148 | 0.148 | 0.148 | 0.148 | 0.148 |
|  | **PII** | 0.074 | 0.074 | 0.074 | 0.074 | 0.074 | 0.074 | 0.074 | 0.074 | 0.074 |
|  | **PDD** | 0.074 | 0.074 | 0.074 | 0.074 | 0.074 | 0.074 | 0.074 | 0.074 | 0.074 |
|  | **PID** | 0.037 | 0.037 | 0.037 | 0.037 | 0.037 | 0.037 | 0.037 | 0.037 | 0.037 |
|  |  | 1.000 | 1.000 | 1.000 | 1.000 | 1.000 | 1.000 | 1.000 | 1.000 | 1.000 |

Inconsistency Index = 0.000

*(7) Compute the matrix of the variables' scores (construct the matrix S):* Matrix S is a matrix that includes the matrix $R_{norm}$ and the vector of criteria weights (w). Table 7 represents a part of this matrix. The first row is include the criteria weights and the rest rows are include the normalized scores of the variables. Constructing this table will help researchers to calculate the overall priority of each variable.

**Table 7.** A part of matrix S.

| $w_i$ | 0.296 | 0.148 | 0.148 | 0.074 | 0.148 | 0.074 | 0.074 | 0.037 |
|---|---|---|---|---|---|---|---|---|
| $R_{norm}$ | DI | II | DD | ID | PDI | PII | PDD | PID |
| V01 | 0.024 | 0.024 | 0.031 | 0.032 | 0.024 | 0.024 | 0.031 | 0.032 |
| V02 | 0.011 | 0.012 | 0.017 | 0.018 | 0.011 | 0.012 | 0.017 | 0.017 |
| V03 | 0.007 | 0.007 | 0.006 | 0.005 | 0.007 | 0.007 | 0.006 | 0.005 |
| V04 | 0.016 | 0.015 | 0.006 | 0.008 | 0.016 | 0.015 | 0.006 | 0.008 |
| V05 | 0.006 | 0.007 | 0.017 | 0.018 | 0.006 | 0.007 | 0.017 | 0.017 |
| : | : | : | : | : | : | : | : | : |
| V45 | 0.016 | 0.017 | 0.030 | 0.029 | 0.016 | 0.017 | 0.030 | 0.029 |

*(8) Calculate the overall priority of each variable:* In order to calculate the overall priority for each variable, we mulitplied matrix $R_{norm}$ on vector $w_i$ ($v = R_{norm} \times w_i$). Table 8 includes the total priority (TP = OPI + OPD), overall priority of influences (OPI = DI + II + PDI + PII), and the overall priority of dependences (OPD = DD + ID + PDD + PID) for all variables. To determine the model's validity (the differences between model results and the realities), we asked the experts to judge the results of the proposed integrated method (AHP-MICMAC).

**Table 8.** The total (TP), overall influences (OPI) and overall dependences (OPD) priorities.

| Var | OPI | OPD | TP * | Var | OPI | OPD | TP | Var | OPI | OPD | TP |
|-----|-----|-----|------|-----|-----|-----|-----|-----|-----|-----|-----|
| V19 | 0.036 | 0.015 | 0.051 | V21 | 0.009 | 0.014 | 0.023 | V07 | 0.013 | 0.005 | 0.018 |
| V17 | 0.022 | 0.021 | 0.044 | V16 | 0.016 | 0.006 | 0.023 | V29 | 0.016 | 0.001 | 0.018 |
| V12 | 0.023 | 0.015 | 0.038 | V28 | 0.015 | 0.008 | 0.022 | V37 | 0.010 | 0.008 | 0.018 |
| V06 | 0.025 | 0.012 | 0.037 | V26 | 0.015 | 0.007 | 0.022 | V33 | 0.011 | 0.004 | 0.015 |
| V30 | 0.026 | 0.009 | 0.035 | V27 | 0.017 | 0.005 | 0.022 | V40 | 0.007 | 0.007 | 0.014 |
| V18 | 0.018 | 0.014 | 0.032 | V10 | 0.018 | 0.004 | 0.022 | V02 | 0.008 | 0.006 | 0.014 |
| V22 | 0.018 | 0.014 | 0.032 | V15 | 0.012 | 0.010 | 0.022 | V38 | 0.008 | 0.005 | 0.013 |
| V43 | 0.025 | 0.007 | 0.032 | V14 | 0.018 | 0.004 | 0.022 | V34 | 0.009 | 0.004 | 0.013 |
| V20 | 0.016 | 0.013 | 0.028 | V13 | 0.017 | 0.005 | 0.021 | V04 | 0.010 | 0.002 | 0.013 |
| V11 | 0.016 | 0.012 | 0.028 | V32 | 0.014 | 0.008 | 0.021 | V41 | 0.006 | 0.006 | 0.012 |
| V25 | 0.019 | 0.009 | 0.028 | V45 | 0.011 | 0.010 | 0.021 | V36 | 0.009 | 0.003 | 0.012 |
| V01 | 0.016 | 0.011 | 0.027 | V42 | 0.016 | 0.005 | 0.020 | V39 | 0.005 | 0.005 | 0.010 |
| V24 | 0.022 | 0.004 | 0.027 | V08 | 0.011 | 0.008 | 0.020 | V05 | 0.004 | 0.006 | 0.010 |
| V44 | 0.024 | 0.001 | 0.024 | V23 | 0.013 | 0.007 | 0.020 | V35 | 0.008 | 0.000 | 0.008 |
| V31 | 0.017 | 0.006 | 0.024 | V09 | 0.013 | 0.006 | 0.019 | V03 | 0.005 | 0.002 | 0.007 |
| Sum | | | | | | | | | 0.667 | 0.333 | 1.000 |

\* sorted by TP column.

## 3. Results and Discussion

### 3.1. The Weights of Various Types of Classifications

In the AHP-MICMAC method, unlike the MICMAC method, various classes of variables do not have the same weights. As matrix $A_{norm}$ shows, the direct influences (DI) and potential indirect dependence (PID), respectively, have had the highest (0.296) and the lowest (0.037) weights among the various types of classifications. Additionally, the weights of II, DD, and PDI classes (0.148) were the same as each other, but their weights were two times more than the ID, PII, and PDD classes (0.074), which have the same weight. Based on Table 8, the sum of the overall priorities of influences (0.667) is two times more than the sum of the overall priorities of dependences (0.333). This means that the experts believe the characteristics of the influences of variables are more important than the characteristics of the dependencies of variables of agricultural development. It is also true for the sum of potential weights (PDI + PII + PDD + PID = 0.667) compared to actual weights (DI + II + DD + ID = 0.333). As is shown in the following section, the application of these weights may change the priority of variables.

### 3.2. The Most Influence and Dependence Variables

Table 6 demonstrates that for an agricultural development system, the most and the least direct and indirect influence variables (both actual and potential) were, respectively, V19 (government policies and programs) and V5 (consumers' access to agricultural products). The influences of government policies and programs on agricultural development have been discussed by other authors [33,34], but the emphasis of this paper is on the type, the weight, and the rank of these influences. After V19 came V30, V06, and V43, in order of increasing influence. The degree of influence of the other variables is represented in Table 6.

Furthermore, the most and the least direct and indirect dependence variables have, in order, been the amount of agricultural production (V17) and topology (V35). The dependence of agricultural production on other factors and variables has been investigated by numerous scholars and organizations [16,35–37]. Table 6 shows that V12 and V19 are the next most important dependent variables that should be considered by planners and policy-makers. The degree of dependency of other variables is mentioned in Table 6.

### 3.3. The Key Variables

Table 9 is sorted in the MICMAC TP column, indicating the key variables of agricultural development based on both methods (MICMAK and AHP-MIKMAC). In the MICMAK method there are eight types of priorities (see Table 6) with the same weights for each variable; if we needed an overall priority, there is no difference between the various types of variable classes. Yet, as previously noted, the priorities of the variables in AHP-MIKMAC are also dependent on the weights of the variable classes. As it can be seen in Table 9, some of the ranks of the OPI, OPD, and TP have changed (for example: V22, V30, V11, and V43). Aside from this, in the MICMAC part of Table 9, there are a number of similar ranks, such as rank 20 and 37 within the TP column. This means that V16 (trade incentives and restrictions) and V32 (agricultural land area) or V33 (agricultural land laws and regulations) and V41 (the international prices of agricultural products) can have similar roles in agricultural development, but in Iran, this is not the case.

**Table 9.** The TP, OPI, and OPD scores and ranks of variables based on MICMAK and AHP-MICMA.

| | MICMAK | | | | | | AHP-MICMAC | | | | | |
|---|---|---|---|---|---|---|---|---|---|---|---|---|
| Variable | OPI | | OPD | | TP * | | OPI | | OPD | | TP | |
| | Score | Rank | Score | Rank | Score | Rank | Score | Rank | Score | Rank | Score | Rank |
| V19 | 2142 | 1 | 1756 | 3 | 3898 | 1 | 0.036 | 1 | 0.015 | 3 | 0.051 | 1 |
| V17 | 1313 | 8 | 2504 | 1 | 3817 | 2 | 0.022 | 8 | 0.021 | 1 | 0.044 | 2 |
| V12 | 1336 | 7 | 1856 | 2 | 3192 | 3 | 0.023 | 6 | 0.015 | 2 | 0.038 | 3 |
| V06 | 1496 | 4 | 1435 | 9 | 2931 | 4 | 0.025 | 3 | 0.012 | 9 | 0.037 | 4 |
| V22 | 1068 | 11 | 1702 | 4 | 2770 | 5 | 0.018 | 11 | 0.014 | 4 | 0.032 | 7 |
| V18 | 1092 | 10 | 1654 | 5 | 2746 | 6 | 0.018 | 10 | 0.014 | 5 | 0.032 | 6 |
| V30 | 1531 | 2 | 1057 | 14 | 2588 | 7 | 0.026 | 2 | 0.009 | 14 | 0.035 | 5 |
| V11 | 958 | 20 | 1442 | 8 | 2400 | 8 | 0.016 | 19 | 0.012 | 8 | 0.028 | 10 |
| V20 | 933 | 23 | 1461 | 7 | 2394 | 9 | 0.016 | 21 | 0.013 | 7 | 0.028 | 9 |
| V43 | 1522 | 3 | 811 | 22 | 2333 | 10 | 0.025 | 4 | 0.007 | 22 | 0.032 | 8 |
| V01 | 962 | 19 | 1272 | 10 | 2234 | 11 | 0.016 | 20 | 0.011 | 10 | 0.027 | 12 |
| V25 | 1130 | 9 | 1096 | 13 | 2226 | 12 | 0.019 | 9 | 0.009 | 13 | 0.028 | 11 |
| V21 | 557 | 36 | 1645 | 6 | 2202 | 13 | 0.009 | 36 | 0.014 | 6 | 0.023 | 16 |
| V15 | 718 | 29 | 1194 | 11 | 1912 | 14 | 0.012 | 29 | 0.010 | 12 | 0.022 | 22 |
| V45 | 675 | 31 | 1187 | 12 | 1862 | 15 | 0.011 | 31 | 0.010 | 11 | 0.021 | 26 |
| V24 | 1341 | 6 | 511 | 35 | 1852 | 16 | 0.022 | 7 | 0.004 | 36 | 0.027 | 13 |
| V31 | 1047 | 14 | 760 | 24 | 1807 | 17 | 0.017 | 14 | 0.006 | 25 | 0.024 | 15 |
| V26 | 935 | 22 | 864 | 20 | 1799 | 18 | 0.015 | 23 | 0.007 | 20 | 0.022 | 19 |
| V28 | 889 | 24 | 905 | 17 | 1794 | 19 | 0.015 | 24 | 0.008 | 18 | 0.022 | 18 |
| V16 | 975 | 18 | 761 | 23 | 1736 | 20 | 0.016 | 17 | 0.006 | 24 | 0.023 | 17 |
| V32 | 817 | 25 | 919 | 16 | 1736 | 20 | 0.014 | 25 | 0.008 | 16 | 0.021 | 25 |
| V08 | 696 | 30 | 1011 | 15 | 1707 | 22 | 0.011 | 30 | 0.008 | 15 | 0.020 | 28 |
| V27 | 1003 | 15 | 614 | 31 | 1617 | 23 | 0.017 | 16 | 0.005 | 30 | 0.022 | 20 |
| V23 | 760 | 28 | 817 | 21 | 1577 | 24 | 0.013 | 28 | 0.007 | 21 | 0.020 | 29 |
| V13 | 987 | 16 | 569 | 33 | 1556 | 25 | 0.017 | 15 | 0.005 | 33 | 0.021 | 24 |
| V10 | 1057 | 12 | 488 | 37 | 1545 | 26 | 0.018 | 13 | 0.004 | 37 | 0.022 | 21 |
| V09 | 766 | 27 | 757 | 25 | 1523 | 27 | 0.013 | 27 | 0.006 | 23 | 0.019 | 30 |
| V14 | 1051 | 13 | 469 | 38 | 1520 | 28 | 0.018 | 12 | 0.004 | 38 | 0.022 | 23 |
| V42 | 949 | 21 | 555 | 34 | 1504 | 29 | 0.016 | 22 | 0.005 | 34 | 0.020 | 27 |
| V37 | 604 | 34 | 887 | 19 | 1491 | 30 | 0.010 | 34 | 0.008 | 17 | 0.018 | 33 |
| V44 | 1410 | 5 | 77 | 44 | 1487 | 31 | 0.024 | 5 | 0.001 | 44 | 0.024 | 14 |
| V07 | 787 | 26 | 618 | 30 | 1405 | 32 | 0.013 | 26 | 0.005 | 31 | 0.018 | 31 |
| V40 | 401 | 41 | 897 | 18 | 1298 | 33 | 0.007 | 41 | 0.007 | 19 | 0.014 | 35 |
| V02 | 473 | 39 | 692 | 27 | 1165 | 34 | 0.008 | 39 | 0.006 | 27 | 0.014 | 36 |
| V29 | 987 | 16 | 170 | 43 | 1157 | 35 | 0.016 | 18 | 0.001 | 43 | 0.018 | 32 |
| V38 | 495 | 38 | 631 | 29 | 1126 | 36 | 0.008 | 38 | 0.005 | 29 | 0.013 | 37 |
| V33 | 675 | 31 | 446 | 39 | 1121 | 37 | 0.011 | 32 | 0.004 | 39 | 0.015 | 34 |
| V41 | 364 | 42 | 757 | 25 | 1121 | 37 | 0.006 | 42 | 0.006 | 26 | 0.012 | 40 |
| V34 | 525 | 37 | 501 | 36 | 1026 | 39 | 0.009 | 37 | 0.004 | 35 | 0.013 | 38 |
| V05 | 258 | 45 | 692 | 27 | 950 | 40 | 0.004 | 45 | 0.006 | 27 | 0.010 | 43 |
| V39 | 315 | 43 | 611 | 32 | 926 | 41 | 0.005 | 43 | 0.005 | 32 | 0.010 | 42 |
| V36 | 561 | 35 | 342 | 40 | 903 | 42 | 0.009 | 35 | 0.003 | 40 | 0.012 | 41 |
| V04 | 607 | 33 | 286 | 41 | 893 | 43 | 0.010 | 33 | 0.002 | 41 | 0.013 | 39 |
| V03 | 276 | 44 | 229 | 42 | 505 | 44 | 0.005 | 44 | 0.002 | 42 | 0.007 | 45 |
| V35 | 471 | 40 | 0 | 45 | 471 | 45 | 0.008 | 40 | 0.000 | 45 | 0.008 | 44 |

\* sorted by TP column of MICMAC method.

In addition, a few variables, such as V44 (disasters: droughts, floods, earthquakes, etc.), V45 (crop insurance) and V15 (storage facilities), changed drastically in their ranking. In recent years, disasters have played an important role in the agricultural sector of developing countries [38,39]. On the other hand, despite the fact that insurance is an appropriate risk management tool in agriculture [40,41], some variables, including "agricultural extension and education" (V31), "water efficiency" (V28), and "farmers' knowledge, awareness, and skills" (V24) are more important. For this reason, crop insurance (V45) is more dependent than these variables. Storage facilities (V15) is also an important variable in developing countries for agricultural development [42,43], but other variables, such as V24, V26, V28, and V31, are more important.

### 3.4. The Influence-Dependence Chart

If we draw the same chart as Figure 4a, in which the horizontal and vertical axis scales, respectively, are OPD and OPI, then we will have a chart that contains five separate sectors (A, B, C, D and E). Each variable is associated with their influential and dependent indicator (OPI and OPD) across the whole System. All of the variables can then be positioned on an influence-dependence chart (Figure 4b). Each identified area in this chart represents a type of the following variables [29]:

*(A) Input variables:* These variables are highly influential and less dependent. They tend to describe the dynamics of the system and the conditions of the other variables. Because of this, they are the first choice when developing different scenarios and strategies. According to Figure 4b, agricultural development in Iran has three input variables: V44, V24, and V43. This means that in order to develop dynamic and sustainable agricultural development in Iran, we must manage disasters, organize farmers, and improve farmers' knowledge, awareness, and skills. Pavelic et al. [44] and Das [39] showed that flood and drought management is very important for agricultural development in Thailand and India. Also, there are many studies that emphasize the importance of human resources and capital (including schooling, training, organizing, and skills) for agricultural development [4,16,45]. Undoubtedly, building the human capital of smallholder farmers can play a critical role in agricultural growth and development.

*(B) Relay or intermediate variables:* These variables are highly influential and highly dependent. Any change will have high flow throughout the rest variables of the system. Figure 4b demonstrates 11 relay variables for agricultural development in Iran. Among these variables V19 (government policies and programs), V17 (the amount of agricultural production), V12 (agricultural products price), and V06 (marketing) are the most important. Our findings were in line with various studies [11,42,46] in other areas.

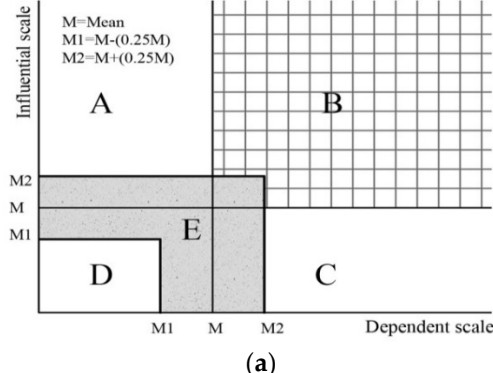 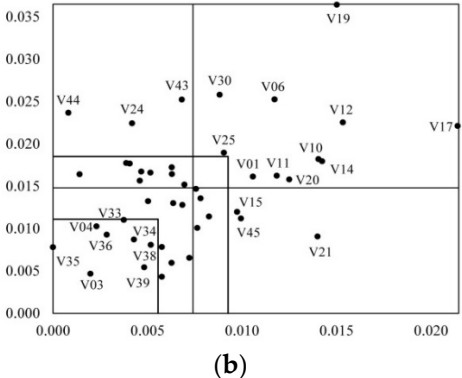

(a)　　　　　　　　　　　　　　(b)

**Figure 4.** The influence-dependence chart. (**a**) The influence-dependence chart area of variables; and (**b**) the position of study variables on the influence-dependence chart.

*(C) Resultant variables:* These variables have a low degree of influence and are highly dependent. Result variables are influenced by both the input variables or determinants (A) and the relay variables (B). Based on Figure 4b, there are three resultant variables in Iran's agriculture system. These are V15

(storage facilities), V21 (farmers' interest and motivation), and V45 (crop insurance). Some studies, such as John and Samuel Noi [33] and Qingshui and Xuewei [22], have also noted the importance of these variables for agricultural development in other areas of the world.

*(D) Excluded or independent variables:* This group is relatively unconnected to the system. They only have a few relationships within it and are neither influential nor dependent variables. Due to their relatively autonomous character and lack of connection to the system, they are not determinants of the future of the system. Therefore, they can be excluded from the next steps of the analysis. As seen in Figure 4b, there are eight excluded variables (V03, V04, V34, V35, V36, V38, and V39) in our study.

*(E) Average variables:* These variables cannot be clearly allocated to the remaining sectors because they are not sufficiently influential or dependent. Though they should be recognized and studied more closely in the future. The remainder of the variables in this study belong to this group.

### 3.5. The Stability and Instability of Agricultural Development System

The pattern of distribution in Figure 4 can not only inform us about the various types of variables, but also presents the stability or instability of a system. The stability of an agricultural system is very important, because agriculture can play an important role in global stability [22,47]. This, then is very helpful to know and informs us about the stability of agriculture system. As Figure 5 shows, if the points are distributed around the main diagonal (see Figure 5a), then the system is unstable. But if the cloud of points is spread along the axis (as L shape: see Figure 5b), it means that the system is stable. The advantage of a stable system is that it introduces a dichotomy between the influential variables, on which one can or cannot act, and the resultant variables which depend on them [27,29]. Based on these explanations and as Figure 5c indicates, the agricultural development system in Iran is unstable. Each variable is both influential and dependent, and any action on one variable has repercussions on all the others and on the original variable. The instability of the Iranian agricultural system has also been highlighted by other studies [48] and in other areas [49–52].

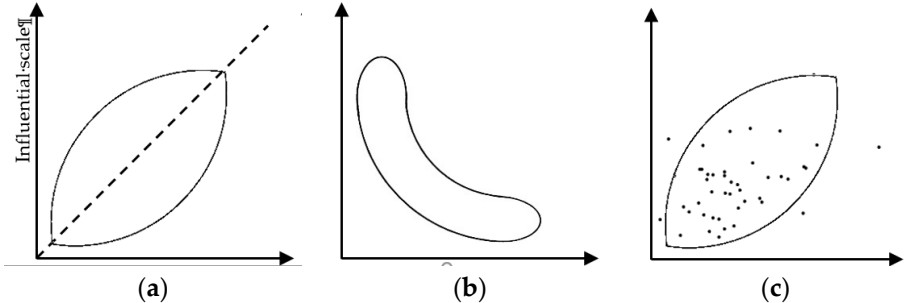

**Figure 5.** System stability according to the influence-dependence chart (**a**) Unstable system; (**b**) Stable system; and (**c**) Iran agricultural development system.

## 4. Conclusions

Agricultural systems, especially in developing countries, are typically complex, and when forming strategies and scenarios, available methods have failed to reveal the essence of such complex systems. Therefore, the main objective of this study was to address this problem by using an integrated method. We integrated the MICMAC and AHP methods, using the MICMAC to determine the various classifications of variables and the AHP method to apply weights to these different variables. The case of the agricultural system of Iran was used to indicate an application of this new integrated method. The results revealed that the various types of variables in agricultural systems, from "actual direct influence" to "potential indirect dependence", did not present similar influences or dependencies on each other. As a result, the ranks of key variables may change by applying the weight of different classification types of variables. Additionally, the AHP-MICMAC method allows us to have a total priority for each variable that helps policy and decision makers to recognize the most important variable according to its dependency and influence on other variables.

For example, in the Iran case, based on the total priority scores of the strategic variables, "farmers' organizing and institutionalizing", "farmers' knowledge, awareness, and skills", and "disasters", respectively, are three main variables that describe the conditions and the dynamics of the other variables of agricultural systems. Therefore, they have a critical role in agricultural growth and development. "Government policies and programs" is the most important intermediate variable for agricultural development. It means the instability of the policies and programs will have high flow throughout the rest variables of the system. "Farmers' interest and motivation", "storage facilities", and "crop insurance" are three main highly dependent variables that are influenced by both input and intermediate variables. There also are some variables, such as "agricultural support system", "water efficiency", "agricultural research", "pricing system", "rural welfare and comforting", "agricultural land area", "transportation and communications", and "trade incentives and restrictions", that they should be recognized and studied more closely in the future.

According to expert opinion, the use of the AHP-MICMAC method has led to a more realistic ranking of the variables and this combination has been able to improve results. It then facilitates the ranking of the variables according to their different types of influences and dependency weights. Without a doubt, any improvement in our understanding of the key variables of a system will lead to forming better scenarios and strategies for development of that system. Although the AHP-MICMAC method is more capable of illustrating the complexities among the variables than many other current methods, it still needs to be developed further so that it can better reflect the interdependency of variables, including economic, social, environmental, religious, etc., which can lead to risky, diverse, and complex agriculture in developing countries, such as Iran. In this regard, performing a study in order to compare the effectiveness of various methods, such as system dynamic modeling, AHP-MICMAC, or cross-impact analysis to display these complexities, is very crucial.

**Author Contributions:** Conceptualization, A.A.B. and H.A., methodology, A.A.B. and M.D.P.; software, A.A.B. and M.Q.; validation, P.L. and H.A.; formal analysis, A.A.B.; investigation, A.A.B. and M.D.P.; resources, A.A.B. and M.D.P.; data curation, A.A.B. and M.Q.; writing—original draft preparation, A.A.B.; writing—review and editing, H.A. and P.L.

**Funding:** This research received no external funding.

**Conflicts of Interest:** The authors declare no conflict of interest.

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
