# Peer review of "Determining Key Agricultural Strategic Factors Using AHP-MICMAC"

_sustainability, doi:10.3390/su11143947_

Round 1
Reviewer 1 Report
In my opinion, the article is valuable because it introduces an integrated
method using MICMAC and AHP techniques to deal with understanding the key
strategic variables of agricultural system. It gives an innovative approach
to understanding complex relationships in the agricultural environment.
However, the chapter Conclusions is lacking a synthetic, brief discussion of
the obtained research results, which would constitute a valuable summary of
the article allowing for making conclusions.
After expanding the Conclusions chapter, because of the interesting
presentation and discussion of the research problem.
Lines 155-164 - I suggest reducing the size of the font (unification with the whole text).
Table 7 - it could be better discussed and explained in the text.
Figure 4 is missing the (b) symbol / sign.
The Conclusions is too general, it should be further elaborated. Please refer in more detail to the obtained research results.
Author Response
Responses to the comments of Reviewer #1
No | The comments of Reviewer | Response | Revisions |
1. | In my opinion, the article is valuable because it introduces an integrated method using MICMAC and AHP techniques to deal with understanding the key strategic variables of agricultural system. It gives an innovative approach to understanding complex relationships in the agricultural environment. However, the chapter Conclusions is lacking a synthetic, brief discussion of the obtained research results, which would constitute a valuable summary of the article allowing for making conclusions.
After expanding the Conclusions chapter, because of the interesting presentation and discussion of the research problem. | Modified: Thanks for your nice feedback. We have tried to address your concern about the “Conclusion section”. In this regard we added a synthetic and brief discussion of the research results as follows: “Agricultural systems, especially in developing countries, are typically complex, and when forming strategies and scenarios, available methods have failed to reveal the essence of such complex systems. Therefore, the main objective of this study was to address this problem by using an integrated method. We integrated the MICMAC and AHP methods, using the MICMAC to determine the various classifications of variables and the AHP method to apply weight to these different variables. The case of the agricultural system of Iran was used to indicate an application of this new integrated method. The results revealed that the various types of variables in agricultural systems, from "actual direct influence" to "potential indirect dependence", did not present similar influences or dependencies on each other. As a result, the ranks of key variables may change by applying the weight of different classification types of variables. Additionally, the AHP-MICMAC method allows us to have a total priority for each variable that helps policy and decision makers to recognize the most important variable according to its dependency and influence on other variables. For example, in Iran case, based on the total priority scores of the strategic variables, “farmers organizing and institutionalizing”, “farmers' knowledge, awareness, and skills” and “disasters” respectively are three main variables that describe the conditions and the dynamics of the other variables of agricultural system. Therefore, they have a critical role in agricultural growth and development. “Government policies and programs” is the most important intermediate variable for agricultural development. It means, the instability of the policies and programs will have high flow throughout the rest variables of the system. “farmers' interest and motivation”, “Storage facilities”, and “crop insurance” are three main high depended variables that are influenced by both input and intermediate variables. There also are some variables, such as “agricultural support system”, “water efficiency”, “agricultural research”, “pricing system”, “rural welfare and comforting”, “agricultural land area”, “transportation and communications”, and “trade incentives and restrictions”, that they should be recognized and studied more closely in the future. According to expert opinion, the use of the AHP-MICMAC method has led to a more realistic ranking of the variables and this combination has been able to improve results. It then facilitates the ranking of the variables according to their different types of influences and dependency weights. Without a doubt, any improvement in our understanding of the key variables of a system will lead to forming better scenarios and strategies for development of that system. Although the AHP-MICMAC method is more capable of illustrating the complexities among the variables than many other current methods, it still needs to be developed further so that it can better reflect the interdependency of variables including economic, social, environmental, religious, etc. which can lead to risky, diverse and complex agriculture in developing countries such as Iran. In this regard, performing a study in order to compare the effectiveness of various methods, such as system dynamic modeling, AHP-MICMAC or cross impact analysis to display these complexities, is very crucial. ” | Please see pages 15-16. |
2. | Lines 155-164 - I suggest reducing the size of the font (unification with the whole text). | Unified: The size of the fonts was reduced. | Please see page 5. |
3. | Table 7 - it could be better discussed and explained in the text. | Discussed: Thanks for your comment. Constructing this Table will help researcher to calculate the overall priority of each variable. We discussed and explained this table in text as follows: “ 7) Compute the matrix of the variables’ scores (construct the matrix S): Matrix S is a matrix that includes the matrix Rnorm and the vector of criteria weights (w). Table (7) represents a part of this matrix. The first row is include the criteria weights and the rest rows are include the normalized scores of the variables. Constructing this Table will help researcher to calculate the overall priority of each variable.” | Please see page 10 |
4. | Figure 4 is missing the (b) symbol / sign. | Modified: Thanks for your comment. Symbol (b) was added to Figure 4.
| Please see page 14 |
5. | The Conclusions is too general, it should be further elaborated. Please refer in more detail to the obtained research results. | Modified: Thanks for your comment. Conclusion section improved as follows: “Agricultural systems, especially in developing countries, are typically complex, and when forming strategies and scenarios, available methods have failed to reveal the essence of such complex systems. Therefore, the main objective of this study was to address this problem by using an integrated method. We integrated the MICMAC and AHP methods, using the MICMAC to determine the various classifications of variables and the AHP method to apply weight to these different variables. The case of the agricultural system of Iran was used to indicate an application of this new integrated method. The results revealed that the various types of variables in agricultural systems, from "actual direct influence" to "potential indirect dependence", did not present similar influences or dependencies on each other. As a result, the ranks of key variables may change by applying the weight of different classification types of variables. Additionally, the AHP-MICMAC method allows us to have a total priority for each variable that helps policy and decision makers to recognize the most important variable according to its dependency and influence on other variables. For example, in Iran case, based on the total priority scores of the strategic variables, “farmers organizing and institutionalizing”, “farmers' knowledge, awareness, and skills” and “disasters” respectively are three main variables that describe the conditions and the dynamics of the other variables of agricultural system. Therefore, they have a critical role in agricultural growth and development. “Government policies and programs” is the most important intermediate variable for agricultural development. It means, the instability of the policies and programs will have high flow throughout the rest variables of the system. “farmers' interest and motivation”, “Storage facilities”, and “crop insurance” are three main high depended variables that are influenced by both input and intermediate variables. There also are some variables, such as “agricultural support system”, “water efficiency”, “agricultural research”, “pricing system”, “rural welfare and comforting”, “agricultural land area”, “transportation and communications”, and “trade incentives and restrictions”, that they should be recognized and studied more closely in the future. According to expert opinion, the use of the AHP-MICMAC method has led to a more realistic ranking of the variables and this combination has been able to improve results. It then facilitates the ranking of the variables according to their different types of influences and dependency weights. Without a doubt, any improvement in our understanding of the key variables of a system will lead to forming better scenarios and strategies for development of that system. Although the AHP-MICMAC method is more capable of illustrating the complexities among the variables than many other current methods, it still needs to be developed further so that it can better reflect the interdependency of variables including economic, social, environmental, religious, etc. which can lead to risky, diverse and complex agriculture in developing countries such as Iran. In this regard, performing a study in order to compare the effectiveness of various methods, such as system dynamic modeling, AHP-MICMAC or cross impact analysis to display these complexities, is very crucial.” |

Reviewer 2 Report
The authors suggest an integrated method that applies both MICMAC and AHP to specify which factors need to be considered for policy making. Then they illustrate the technique using the case of Iran and the development of their agriculture.
I find the issue on determining strategic factors for agriculture development interesting and I think that combining both MICMAC and AHP gives a more specific insight into the ranking of key variables.
In my opinion, the paper in its current form is well written. If anything, I would suggest the authors including some information on the advantages and limitations (potential subjective nature, for example) of both methods.
Author Response
Responses to the comments of Reviewer #2
No | The comments of Reviewer | Response | Revisions |
1. | In my opinion, the paper in its current form is well written. If anything, I would suggest the authors including some information on the advantages and limitations (potential subjective nature, for example) of both methods. | Corrected: Thanks to your nice comment, we added some more information about the advantages and limitations of these method as follows: “Each of these methods alone has advantages and limitations for example MICMAC can investigate multiple variables at the same time, but it does not give an overall priority score for each variable. On other side, AHP considers only direct impact of variables, but it gives and overall priority score for each variable. This study has tried to overcome these constraints and to consider their advantages by combining them and proposing an integrated method. It is our hope that this new integrated method will supply instructions for the development of agriculture, and find wider applications in complex systems.” | Please see page 2. |
